# Hyperedge Anomaly Detection with Hypergraph Neural Network

**Md. Tanvir Alam**                                                          *tanvir15@du.ac.bd*
*Department of Computer Science and Engineering*
*University of Dhaka, Bangladesh*

**Md Mahmudur Rahman**                                             *mahmudur@cse.du.ac.bd*
*Department of Computer Science and Engineering*
*University of Dhaka, Bangladesh*

**Md. Fahim Arefin**                                                     *fahim@cse.du.ac.bd*
*Department of Computer Science and Engineering*
*University of Dhaka, Bangladesh*

**Chowdhury Farhan Ahmed**                                            *farhan@du.ac.bd*
*Department of Computer Science and Engineering*
*University of Dhaka, Bangladesh*

**Zisan Mahmud**                                          *zisan-2020815633@cs.du.ac.bd*
*Department of Computer Science and Engineering*
*University of Dhaka, Bangladesh*

**Md. Sadman Sakib**                                   *mdsadman-2020015659@cs.du.ac.bd*
*Department of Computer Science and Engineering*
*University of Dhaka, Bangladesh*

**Carson K. Leung**                                            *Carson.Leung@UManitoba.ca*
*Department of Computer Science*
*University of Manitoba, Canada*

**Reviewed on OpenReview:** *https://openreview.net/forum?id=etPYIk1BqO*

## Abstract

Hypergraph is a data structure that enables us to model higher-order associations among data entities. Conventional graph-structured data can represent pairwise relationships only, whereas hypergraph enables us to associate any number of entities, which is essential in many real-life applications. Hypergraph learning algorithms have been well-studied for numerous problem settings, such as node classification, link prediction, etc. However, much less research has been conducted on anomaly detection from hypergraphs. Anomaly detection identifies events that deviate from the usual pattern and can be applied to hypergraphs to detect unusual higher-order associations. In this work, we propose an end-to-end hypergraph neural network-based model for identifying anomalous associations in a hypergraph. Our proposed algorithm operates in an unsupervised manner without requiring any labeled data. Extensive experimentation on several real-life datasets demonstrates the effectiveness of our model in detecting anomalous hyperedges.

# 1   Introduction

Graph-structured data has the natural ability to present pairwise relationships between data entities. It helps to model various real-life problems across various domains, such as social networks, biological systems, and recommendation systems. Therefore, the prevalence of graph-structured data in real-life applications has driven growing attention to developing graph learning algorithms. In addition, graph neural network-based machine learning models have been explored extensively for node classification, link prediction, anomaly detection, etc., as these models have overcome the limitations of traditional learning algorithms to capture complex pairwise relationships from non-Euclidean data. However, we need to associate more than two entities in many scenarios, such as co-authorship networks, social groups, etc. Graphs fail to preserve such relationships beyond pairs and are limited to modeling relationships between a pair of entities. On the other hand, hypergraphs can model higher-order complex relationships and associate any number of entities, overcoming the limit. In order to utilize the signals of higher-order complex relationships, neural network-based machine learning models have been developed for hypergraphs. These models primarily focus on node classification, node clustering, link prediction, etc. Comparatively, anomaly identification in hypergraphs has received less attention.

Anomaly detection is a core data mining task that identifies unusual events deviating from the norm. Graph-based anomaly detection has attracted researchers' attention due to the scope of exploring higher-order complex relationships. Methods have been devised to detect anomalies within a single graph (Noble & Cook, 2003; Sun et al., 2005; Akoglu et al., 2010; Eswaran et al., 2018; Chen & Sun, 2020). The earlier researchers focused on the use of statistical models or substructure mining, which limits the capability of the methods in terms of scalability and generalization. Later researchers have utilized the expressive power of Graph Neural Networks (GNN) for node classification (Grover & Leskovec, 2016), clustering (Tsitsulin et al., 2023), link prediction (Zhang & Chen, 2018), etc. Consequently, GNNs are applied in the work of Wang et el. (Wang et al., 2021) for graph anomaly detection. Moreover, some research on developing algorithms to detect anomalous nodes within a graph (Dou et al., 2020; Tang et al., 2022), with applications such as identifying compromised nodes in a network, detecting spam accounts in social networks, etc. To spot unusual connections between nodes (Ranshous et al., 2016), uncovering unusual communication patterns or fraudulent transactions (Zhang et al., 2022a), anomalous edge detection becomes very important. On the contrary, graph-level anomaly detection identifies anomalous graphs within a set of graphs (Qiu et al., 2022; Zhang et al., 2022b).

The ability to preserve multi-entity relationships makes hypergraphs a convenient choice for modeling many real-life problems, which opens the door to research on anomaly identification from hypergraphs, aiming to detect uncommon higher-order associations represented by hyperedges. The research on hyperedge anomaly detection has been used for many practical applications, such as identifying abnormal network traffic patterns involving multiple devices, unusual group activities in social networks, fraudulent financial transactions involving numerous parties, and rare interactions among multiple genes. Moreover, the large input space, resulting from the combinatorial nature of hyperedges, presents a fundamental challenge in learning hyperedge-related tasks, in contrast to node-related tasks. It necessitates designing a model both efficient and scalable, which can implicitly capture higher-order relationships without exhaustively enumerating or storing all possible hyperedges. Nowadays, the research on hyperedge anomaly detection has focused mainly on hashing and statistical similarity measures. For example, the LSH (Ranshous et al., 2018) devised a similarity-measure-based anomaly detection method for hyperedge streams using *minhash* and *locally sensitive hashing*. In addition, the HashNWalk (Lee et al., 2022) employs *random walk* similarities and the hash function to determine anomalies in hyperedge streams. Moreover, Silva & Willett (2008) have tailored a variational *expectation-minimization* algorithm for hypergraphs to detect anomalies. But none of these approaches have integrated node feature information into the process, which limits their *generalizability* and *effectiveness.*

Similar to graph neural networks, hypergraph neural networks have proven effective in extracting expressive representations. Research like HGNN (Feng et al., 2019), HyperGCN (Yadati et al., 2019), and AllSet (Chien et al., 2022) has introduced different convolution or message-passing-based frameworks for hypergraph neural networks and has proven to be effective in hypergraph node classification. On the other hand, HCoN (Wu

et al., 2023) has developed a model for both node and hyperedge classification. In addition, NHP (Yadati et al., 2020) has utilized the graph convolutional network for link prediction in hypergraphs. Moreover, AHP (Hwang et al., 2022) has adopted adversarial training to a hypergraph neural network for hyperedge prediction. However, the potential of hypergraph neural networks for anomaly detection is still unexplored. To the best of our knowledge, no effective deep neural network models have been proposed for anomaly detection from hypergraphs in the literature.

Considering the research gaps, in this work, we design a novel hypergraph neural network-based end-to-end model to address the problem of detecting anomalous hyperedges. Therefore, we propose a Hyperedge Anomaly Detection (HYPADE) algorithm that exploits the attributes or characteristics associated with the hypergraph nodes. HYPADE is an unsupervised model that operates without requiring any prior information about the data distribution of anomalies, as well as any labeled anomalies, which are rare in the domain of anomaly detection. The main contributions in this work can be summarized as–

- We devise HYPADE, a novel end-to-end hypergraph neural network model, to detect anomalous hyperedges.

- Our proposed HYPADE model operates in an unsupervised manner without requiring labeled anomalies.

- We curate six real-life and six synthetic hypergraph datasets from different domains to evaluate the performance of HYPADE.

- We conduct extensive experimentation on both real-life and synthetic hypergraphs to demonstrate the effectiveness of our method in terms of standard measures.

The rest of the paper is organized as follows: Section 2 discusses the related works, Section 3 defines the problem, and Section 4 describes the proposed model. Section 5 presents the experimental settings and result analysis. Finally, we conclude the paper in Section 6.

## 2 Related works

In this section, we discuss the research works related to our work. First, we review state-of-the-art hypergraph learning methods in Section 2.1. Then, we focus on existing graph anomaly detection algorithms in Section 2.2 and hypergraph anomaly detection methods in Section 2.3.

### 2.1 Hypergraph Learning Methods

Hypergraph neural networks have been devised for learning hypergraph-related tasks due to the impactful utilization of deep learning to graph-structured data. Initially, a spectral hypergraph embedding method based on the hypergraph Laplacian was introduced by Zhou et al. (Zhou et al., 2006). Later, HGNN (Feng et al., 2019) has generalized the graph convolutional network to the hypergraph domain by propagating features through a single-stage message-passing framework. In addition, HyperGCN (Yadati et al., 2019) has also adapted the graph convolutional networks to hypergraphs by approximating the hypergraph with a graph. Moreover, to properly capture the higher-order relationships in the representations, AllSet (Chien et al., 2022) has proposed a two-stage message-passing framework. In this approach, instead of learning the node representations from the neighborhood, the hyperedge representations are learned from the node representations (features) of the previous layer first. Then, the hyperedge representations are aggregated to learn the node representations. Recently, HCoN(Wu et al., 2023) has devised a model for both node and hyperedge classification that considers both node and hyperedge from the previous layer. Hwang et al. (Hwang et al., 2022) has explored a method for hyperedge prediction based on hypergraph neural networks that adopts generative adversarial training to generate negative examples.

## 2.2 Graph Anomaly Detection

Anomaly detection methods for graph-structured data can be broadly categorized into four categories: (1) Node Anomaly Detection, (2) Edge Anomaly Detection, (3) Subgraph Anomaly Detection, and (4) Graph-level Anomaly Detection. OCGNN (Wang et al., 2021) has proposed a one-class graph neural network for node anomaly detection utilizing the expressive capability of graph neural networks. By extending a one-class support vector machine, it learns a hypersphere with a center vector and a radius where normal nodes are mapped to embeddings confined within the defined hypersphere. Then, the nodes whose embeddings lie beyond the hypersphere are considered anomalous. Moreover, BWGNN (Tang et al., 2022) has analyzed spectral energy distributions and devised a graph neural network to detect node anomalies. In addition, a scalable approach for detecting anomalies in a dynamic graph setting has been proposed by Ranshous et al.(Ranshous et al., 2016). Importantly, IGAD (Zhang et al., 2022b) has introduced a point mutual information-based loss function to graph neural networks for graph-level anomaly detection. In addition, OCGTL (Qiu et al., 2022) develops a self-supervised one-class graph transformation model that learns an anomaly score function to detect anomalies from a set of graphs.

## 2.3 Hypergraph Anomaly Detection

For hypergraph data, research on anomaly detection has mostly focused on hyperedge-level anomalies. An anomaly detection method for hyperedge streams, using *minhash* and *locally sensitive hashing*, has been developed in LSH (Ranshous et al., 2018). On the other hand, HashNWalk (Lee et al., 2022), an incremental algorithm, has used random walk similarities and hash functions to identify anomalies in hyperedge streams. To scale the algorithm for large-scale streams, it maintains a constant-size summary of the stream to calculate the anomaly scores. In addition, a variational *expectation-maximization* algorithm has been tailored for hypergraphs to detect anomalies through probability mass function estimation in (Silva & Willett, 2008). However, none of these methods leverage the expressive power of hypergraph neural networks, which have been proven to be effective for other data mining tasks such as node classification, link prediction, and others.

# 3 Preliminaries

In this section, we introduce the notations and preliminary concepts related to hyperedge anomaly detection in Section 3.1. Later, we formally define the hyperedge anomaly detection problem in Section 3.2.

Table 1: Summary of notations.

| Notation | Description |
|---|---|
| $H = (V, E, X)$ | A hypergraph |
| $V = \{v_1, v_2, \ldots, v_{|V|}\}$ | Set of nodes or vertices |
| $E = \{e_1, e_2, \ldots, e_{|E|}\}$ | Set of hyperedges |
| $X \in \mathbb{R}^{|V| \times d}$ | Node feature matrix ($d$-dimensional features) |
| $e \subseteq V$ | Each hyperedge is a subset of the node set |
| $A_H \in \mathbb{R}^{|E| \times |V|}$ | Hypergraph incidence matrix |
| $A_H(i, j) = 1$ | Vertex $v_j$ is in hyperedge $e_i$ |
| $X_v \in \mathbb{R}^d$ | Feature vector of node $v$ |

## 3.1 Notations

A hypergraph can be represented as $H = (V, E, X)$, where $V = \{v_1, v_2, ..., v_{|V|}\}$ is the set of nodes or vertices, $E = \{e_1, e_2, ..., e_{|E|}\}$ is the set of hyperedges, and $X \in \mathbb{R}^{|V| \times d}$ is the feature matrix. Each hyperedge is a subset of the vertices set that it connects, i.e., $\forall_{e \in E} e \subseteq V$. The incidence matrix of hypergraph $H$ is denoted by $A_H \in \mathbb{R}^{|E| \times |V|}$, where the $(i, j)$-th entry is 1 if the $i$-th hyperedge contains the $j$-th vertex, and 0

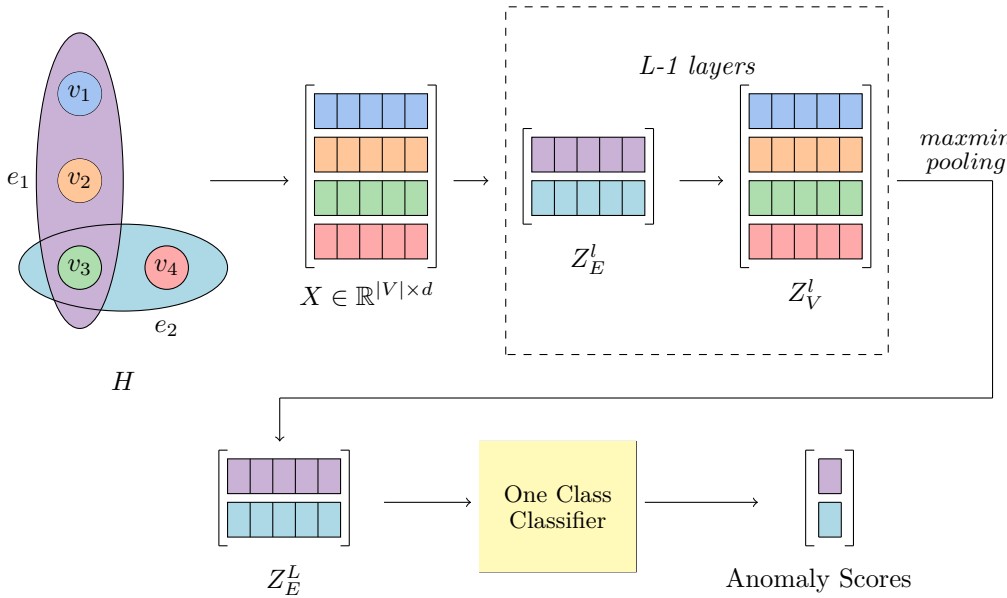

Figure 1: Flow Diagram of the Proposed Method

otherwise. $X_v$ represents the $d$-dimensional feature vector of the node $v$. In Table 1, we present a summary of the notations used to define and describe the hypergraph structure and its components.

### 3.2 Problem Definition

Given a hypergraph $H = (V, E, X)$, the task of hyperedge anomaly detection is to learn a function $f : 2^V \to [0, 1]$ that assigns an anomaly score to a hyperedge. A higher score indicates a higher likelihood of being an anomaly for a hyperedge. Note that the hyperedge is not necessarily a member of E, which also provides us the flexibility of predicting anomaly scores for unseen hyperedges.

## 4 Proposed Methods

In this section, we present a hyperedge anomaly detection model, HYPADE. Mainly, the process learns the node embeddings using a two-stage hypergraph neural network. For each hyperedge, at first, it derives its embedding by pooling from the embeddings of the nodes that it connects. Then, in the subsequent layer, for each node, it derives its embedding by pooling from the embeddings of the hyperedges that contain it. Later, it derives the hyperedge embeddings by applying $maxmin$ pooling to the node embeddings. We define the centroid of the hypergraph as the mean of all the hyperedge embeddings in the hypergraph. Finally, we train a one-class classifier that optimizes the mean Euclidean distance between the hyperedge embeddings and the centroid.

In Figure 1, a snapshot of the process is demonstrated. In the figure, the hypergraph $H$ contains four vertices: $v_1$, $v_2$, $v_3$, and $v_4$. Moreover, two hyperedges are $e_1$ and $e_2$ where $e_1$ associates the vertices $v_1$, $v_2$, $v_3$ and $e_2$ associates the vertices $v_3$, $v_4$. There, the hyperedge embeddings $Z_E^l$ are learned by aggregating the features from the matrix $X$, and then the node embeddings $Z_V^l$ are derived. At the final level, the hyperedge embeddings are learned by applying $maxmin$ pooling to the node embeddings. Lastly, a one-class classifier is applied to find the anomaly scores. The detailed process is described in the following sections.

### 4.1 Learning Node Embeddings

In the proposed model, it begins by learning the node embeddings of the hypergraph using a hypergraph neural network architecture. The vector representation of a hyperedge $e \in E$ at layer $l$, $Z_e^l$, is derived from the embeddings of the nodes it contains from the previous layer.

$$Z_e^l = ENN^l \left( \sum_{v \in e} Z_v^{l-1} \right) \tag{1}$$

Here, $ENN^l$ is the multi-layer perceptron for hyperedges at layer $l$. $Z_v^{l-1}$ is the node embedding of the node $v$ of layer $l-1$. We derive the vector representation of a node $v \in V$ at layer $l$, denoted as $Z_v^l$, from the embeddings of the hyperedges containing the node.

$$Z_v^l = VNN^l \left( \sum_{e \in E_v} Z_e^l \right) \tag{2}$$

Here, $VNN^l$ is the multi-layer perceptron for nodes at layer l. $E_v$ is the set of hyperedges that contains $v$. That is, $E_v = \{e : e \in E \text{ and } v \in e\}$. Note that $Z_v^0 = VNN^0(X_v)$, where $X_v$ is the $d$-dimensional feature vector of the node $v$ from the feature matrix.

### 4.2 Learning Hyperedge Embeddings

The embedding of a hyperedge $e$ at the final layer $L$, $Z_e^L$, is learned by pooling from embeddings of the nodes it connects. The process uses *maxmin* pooling, subtracting the element-wise minimum values from the element-wise maximum values, which captures the diversity of the nodes within the hyperedge. This information about diversity within a hyperedge may be crucial for detecting anomalies.

$$Z_e^L = \max \left\{ \bigcup_{v \in e} Z_v^{L-1} \right\} - \min \left\{ \bigcup_{v \in e} Z_v^{L-1} \right\} \tag{3}$$

### 4.3 One Class Classifier

To make the proposed model trainable end-to-end, a one-class classifier has been developed by optimizing an objective function. Then, a centroid of the given hypergraph is computed as a reference point for calculating anomaly scores. The centroid, $C_H$, is calculated by taking the mean of all the hyperedge embeddings in the hypergraph.

$$C_H = \frac{1}{|E|} \sum_{e \in E} (Z_e^L) \tag{4}$$

The anomaly score $f(e)$ of a hyperedge $e$ is calculated by its Euclidean distance from the embedding $Z_e^L$ to the hypergraph centroid $C_H$.

$$f(e) = \|Z_e^L - C_H\|_2 \tag{5}$$

*Objective Function*: Considering the hyperedges in $E$ as inliers, the objective of the process is to minimize their anomaly scores. The proposed model trains the one-class classifier by minimizing the mean anomaly score over all the hyperedges in the hypergraph. Thus, the objective function is defined as follows:

$$\min \frac{1}{|E|} \sum_{e \in E} \|Z_e^L - C_H\|_2 \tag{6}$$

In the proposed approach, rather than fixing the centroid $C_H$ as done in existing one-class classifiers (Ruff et al., 2018), it dynamically updates the centroid based on the changing values of $Z_e^L$ for $e \in E$ during training. However, dynamically updating the centroid makes the naive solution of setting all the weights to zero the most optimized solution. Here, the embeddings of all the hyperedges and the centroid become zero vectors, and the model fails to learn anything from the data. This issue is discussed in (Ruff et al., 2018) as the *"Hypersphere Collapse"* issue. To prevent the *"Hypersphere Collapse"* issue, a hyperparameter called *loss_threshold* is introduced. When the loss value falls below *loss_threshold*, it stops further optimization to prevent hypersphere collapse.

The pseudocode of the training phase of HYPADE is presented in Algorithm 1. In line 1, the initial node embeddings are calculated. Then, it learns the layer-wise hyperedge and node embeddings in lines 5 and 6, respectively. In line 9, it calculates the hyperedge embeddings of the final layer by applying the *maxmin* function. After computing the centroid, $C_H$, and *loss* in lines 11 and 12, it updates the parameters of the multi-layer perceptron in line 15.

---

**Algorithm 1:** HYPADE Training

**Input** : $H = (V, E, X)$: A hypergraph, *loss_threshold*: The loss threshold
**Output:** $Z_V^{L-1}$: The node embeddings, $C_H$: The centroid of the hypergraph $H$

1 **begin**
2    **while** *True* **do**
3      $Z_V^0 \leftarrow VNN^0(X)$;
4      **for** $l \leftarrow 1 \rightarrow L - 1$ **do**
5        $Z_E^l \leftarrow ENN^l(A_H Z_V^{l-1})$;
6        $Z_V^l \leftarrow VNN^l(A_H^T Z_E^l)$;
7      **end**
8      **for** $e \in E$ **do**
9        $Z_e^L \leftarrow \max\{\bigcup_{v \in e} Z_v^{L-1}\} - \min\{\bigcup_{v \in e} Z_v^{L-1}\}$;
10      **end**
11      $C_H \leftarrow \frac{1}{|E|} \sum_{e \in E}(Z_e^L)$;
12      $loss \leftarrow \frac{1}{|E|} \sum_{e \in E} \|Z_e^L - C_H\|_2$;
13      **if** $loss \leq loss\_threshold$ **then**
14        Break;
15      Update the parameters of $\bigcup_{i=0}^{L-1} VNN_i$ and $\bigcup_{i=1}^{L} ENN_i$ to minimize *loss*;
16    **end**
17 **end**

---

In Algorithm 2, the pseudocode of the function to calculate the anomaly score of a given hyperedge is presented. It computes the hyperedge embedding by applying the *maxmin* function to the final embeddings of the nodes that the hyperedge contains and then returns the distance from the centroid as the anomaly score.

In summary, through the iterative message passing and aggregation processes of the hypergraph neural network from nodes to hyperedges and vice versa, this model effectively propagates information across multiple hops, enabling the utilization of global context and long-range dependencies. In the final layer, it leverages a *max-min* pooling technique to compute the hyperedge embedding that captures the diversity of the constituent nodes. From the embeddings of the inlier hyperedges, it trains a one-class classifier that attempts to map the embeddings closer. Then, based on the deviation of the embeddings from the expectation, it assigns anomaly scores to hyperedges.

## 5 Experimental Results

In this section, the experimental settings and the performance of an extensive result analysis of the proposed algorithm are presented. The rest of the parts are organized as Section 5.1 describes the experimental setup.

---

**Algorithm 2:** Anomaly Score Prediction

---

**Input** : $e \subseteq V$: A hyperedge, $H = (V, E, X)$: A hypergraph, $C_H$: The centroid of the hypergraph $H$,
$Z_V^{L-1}$: The node embeddings

**Output:** *score*: the anomaly score of hyperedge $e$

**1 begin**

**2**    $Z_e^L \leftarrow \max\{\bigcup_{v \in e} Z_v^{L-1}\} - \min\{\bigcup_{v \in e} Z_v^{L-1}\}$;

**3**    $score = \|Z_e^L - C_H\|_2$;

**4 end**

---

Moreover, Section 5.2 presents the details of the datasets. In addition, Section 5.3 discusses the baseline considered for comparison, and Section 5.4 analyzes the experimental results. Finally, in Section 5.5, we visualize the anomaly scores for inlier and anomalous hyperedges. The detailed experiments on synthetic datasets, including dataset descriptions, results, loss analysis, and anomaly score visualizations, are provided in Appendix A.

## 5.1 Experimental Setup

To evaluate the effectiveness of HYPADE, we conducted experiments on six real-world datasets and six synthetic datasets. The model was implemented in Python using the PyTorch framework and executed on a machine equipped with an Intel Core i7-6700K CPU running at 4.00 GHz and 16 GB RAM. For each dataset, we performed five-fold cross-validation. In each fold, the inlier hyperedges were partitioned into 80% training and 20% testing subsets. Only the training inlier hyperedges were used to optimize the model parameters. During evaluation, the test set consisted of the held-out inlier hyperedges and the anomalous hyperedges. To mitigate the effect of class imbalance, the inlier test hyperedges were oversampled to match the number of anomalous hyperedges. The reported results correspond to the mean AUROC score across the five folds. The proposed architecture consists of two node encoders and two hyperedge encoders, each implemented as a fully connected layer followed by a ReLU activation function. The initial node encoder projects the input node features to a 128-dimensional hidden space, while the subsequent encoders operate in an 8-dimensional latent space. Hyperedge representations are computed using the proposed max-min pooling strategy, and anomaly scores are calculated as the Euclidean distance between hyperedge embeddings and the dynamically updated centroid. The model parameters were optimized using the Adam optimizer with a learning rate of $10^{-4}$. The number of message-passing layers was set to $L = 2$ for all datasets. Training was terminated when the objective value fell below the predefined loss threshold of $10^{-4}$, which was introduced to prevent hypersphere collapse during optimization, or when the maximum number of training epochs was reached. Following the unsupervised anomaly detection setting, no validation set was used, and labeled anomalous hyperedges were utilized exclusively for performance evaluation.

## 5.2 Datasets

To perform our experiments, we utilized six real-world hypergraph datasets from diverse domains and generated an additional six synthetic datasets. A detailed description of the real-world datasets is provided in Section 5.2.1, while the characteristics of the synthetic datasets are discussed in Appendix A.1.

### 5.2.1 Real-life Datasets

We perform our experiments on six real-world benchmark datasets drawn from diverse domains: Mushroom (mus, 1981), CiteSeer (Sen et al., 2008), CoraA (Yadati et al., 2019), Cora (Sen et al., 2008), PubMed (Namata et al., 2012), and DBLP (Yadati et al., 2019). The mushroom dataset contains information about various mushroom species. For each species, 22 nominal value attributes are recorded. The species are categorized as either edible or poisonous. We created a hypergraph using the nominal values as nodes, with each hyperedge representing a species connecting the nodes of its nominal values. The edible species are considered inliers, whereas the poisonous species are considered anomalies. Due to the absence of any node features, we assigned each node a unique identity vector as a feature.

Table 2: Statistics of the real-life datasets.

| Dataset | Mushroom | CiteSeer | CoraA | Cora | PubMed | DBLP |
|---|---|---|---|---|---|---|
| Number of nodes, \|V\| | 117 | 1,458 | 2,388 | 1,434 | 3,840 | 41,302 |
| Number of hyperedges, \|E\| | 8,124 | 1,079 | 1,072 | 1,579 | 7,963 | 22,363 |
| Number of train hyperedges | 3,133 | 242 | 350 | 272 | 3,604 | 2,897 |
| Number of test hyperedges | 8,416 | 1,554 | 1,270 | 2,480 | 6,916 | 37,484 |
| Average hyperedge size | 22.00 | 3.20 | 4.27 | 3.03 | 4.34 | 4.45 |
| Maximum hyperedge size | 23 | 27 | 44 | 6 | 172 | 203 |

Table 3: AUROC score (%) of our method and baseline methods on real-life datasets.

| Dataset | Mushroom | CiteSeer | CoraA | Cora | PubMed | DBLP |
|---|---|---|---|---|---|---|
| LSH | 32.02±1.43 | 63.09±2.14 | 46.25±1.76 | 50.40±1.95 | 53.35±1.62 | 48.45±2.08 |
| HashNWalk | 57.47±1.88 | 48.63±2.21 | 50.07±1.93 | 52.57±2.02 | 59.04±1.84 | 53.65±1.97 |
| VEM | 72.04±1.52 | 52.20±1.87 | 50.76±2.11 | 55.39±1.76 | 54.07±1.58 | 50.18±1.91 |
| HGNN | 87.72±0.94 | 36.84±1.36 | 29.41±1.48 | 42.17±1.24 | 55.83±1.67 | 25.96±1.72 |
| HYPADE-Mean | **100.00**±0.00 | 54.24±1.65 | 39.83±1.42 | 65.72±1.18 | 36.37±1.96 | 46.08±1.75 |
| HYPADE-Fixed | 50.01±1.74 | 29.53±1.37 | 29.71±1.21 | 33.12±1.54 | 57.07±1.63 | 16.51±1.28 |
| HYPADE | **100.00**±0.00 | **71.56**±1.12 | **64.70**±1.05 | **66.31**±0.93 | **59.07**±1.26 | **56.58**±1.11 |

CiteSeer, Cora, and PubMed are three co-citation datasets containing information about papers, their citations, and co-citations. In the hypergraph representation, Each node represents a paper, and each hyperedge connects papers cited in another paper. Bag-of-words features from the paper abstracts are used as node features. For hyperedge classification, on datasets where labels are unavailable, the labels of the nodes in a hyperedge are used to label the hyperedge (Wu et al., 2023). Following this approach, we also utilized the node labels. We considered the hyperedges containing a node labelled with the most frequent node label as inliers and all the other hyperedges as anomalies.

CoraA and DBLP are two co-authorship datasets we considered in our experiments. Each node represents a paper in these hypergraphs, and the hyperedges connect papers authored by a specific author. Similar to co-citation datasets, bag-of-words features of the abstracts of the corresponding papers are used as node features. The same strategy as co-citation datasets is used to distinguish the inliers and anomalous hyperedges. In Table 2, we present the statistical summary of all six real-life datasets, including the dataset description.

Since publicly available benchmark datasets with ground-truth hyperedge anomaly annotations are scarce, we defined anomaly labels using a proxy-labeling strategy. We note that these labels should be regarded as proxy anomaly labels rather than naturally occurring anomaly annotations. While this provides a reproducible benchmark in the absence of standard hyperedge anomaly datasets, class membership does not necessarily correspond to real-world anomalous behavior and therefore constitutes a limitation of the current setup.

We also generated six synthetic datasets for our experiments; their descriptions and characteristics are provided in Appendix A.1.

### 5.3 Baselines

Our experiments considered three baseline methods for performing comparative performance analysis. We applied LSH (Ranshous et al., 2018), an algorithm proposed for anomaly detection in a hyperedge stream, by randomly shuffling the order of hyperedges. Similarly, we have considered HashNWalk (Lee et al., 2022), another anomaly detection algorithm for hyperedge streams. We implemented VEM (Silva & Willett, 2008), a variational expectation-minimization algorithm, to detect hyperedge anomalies. To calculate the AUROC score, we first normalized the anomaly score for all the algorithms. We also considered HGNN (Feng et al., 2019) to learn node representations and applied the same one-class anomaly detection objective used in

HYPADE on the resulting hyperedge representations. Furthermore, we have implemented two variations of our algorithm. First, we utilized mean pooling instead of $maxmin$ pooling (Equation (3)) in the final layer to examine the significance of capturing diversity in detecting anomalies and named the model HYPADE-Mean. Second, we fixed the centroid $C_H$ as proposed in existing one-class classifiers(Ruff et al., 2018) and named the model HYPADE-Fixed. However, for HYPADE-Fixed, the $loss$ value (Algorithm 1, line 13) converges below the $loss\_threshold$ value of 0.0001 as used in our experimental setup. We have run the training for 1000 epochs to deal with the issue instead of using the $loss\_threshold$ value.

## 5.4   Results

In Table 3, we present the AUROC scores in percentages of our model along with the baselines for the six real-life datasets. The best result of all models is highlighted in bold. We observe that our model has outperformed all the baselines significantly on all the datasets. On dataset Mushroom, HYPADE-Mean and HYPADE both achieve perfect scores of 100%, significantly outperforming other methods.

Among the baseline methods, HGNN achieves a strong AUROC score of 87.72% on Mushroom, outperforming the traditional hypergraph anomaly detection methods. However, its performance is considerably lower on the citation and co-authorship datasets, where it is consistently outperformed by HYPADE. This suggests that simply combining a hypergraph neural encoder with a one-class objective is insufficient, and highlights the importance of the proposed maxmin pooling and dynamic centroid optimization mechanisms. VEM is the next best, with a score of 72.04%. The AUROC score for HashNWalk is 57.47%, while LSH performs poorly at 32.02%. The AUROC score of our model is 27.96% more than that of VEM, the highest among the baselines. On CiteSeer, the AUROC score is 8.49% more than LSH, the best-performing baseline. The improvements are also significant for CoraA and Cora, which are 13.94% and 10.92%, respectively. On PubMed, the difference is marginal compared to HashNWalk (0.03%). For the hypergraph DBLP, the improvement is 2.92%. The better performance is also evident in the comparison with the variants HYPADE-Mean and HYPADE-Fixed. HYPADE-Fixed performs the worst among the HYPADE variants, particularly on datasets Mushroom, CiteSeer, CoraA, and DBLP. HYPADE-Mean performs well on certain datasets, particularly Mushroom and Cora, but fails to match the overall performance of HYPADE. The better AUROC score of our proposed model than HYPADE-Mean demonstrates the importance of the knowledge of diversity within a hyperedge for anomaly detection. The AUROC score of HYPADE-Fixed is also lower than our proposed model. Compared to HYPADE-Mean, the AUROC score is lower on all the datasets except PubMed for HYPADE-Fixed. When the centroid is fixed, it becomes challenging for the model to converge and learn due to the possibility of selecting a poor centroid. Having a dynamic centroid allows the model to learn more easily by choosing a suitable centroid, leading to improved performance.

The detailed results and analysis on the synthetic datasets are provided in Appendix A.2.

In Figure 2, we present the loss values over the first 1,000 epochs for the real-life datasets. We compare two methods, HYPADE-Fixed and our proposed HYPADE algorithm, to demonstrate the effect of updating the centroid dynamically. For both methods, we started the training with the same set of parameters initialized randomly and recorded the loss values over epochs. For all the datasets, the loss value decreases sharply in the initial epochs and gradually stabilizes. The decrement in loss value is comparatively larger for HYPADE than HYPADE-Fixed, and so the model converges fast and results in better performance. After the initial epochs, although having a higher loss value, the HYPADE-Fixed model fails to minimize the loss value significantly. For dataset Mushroom (Figure 2-a, the loss value starts at 74.63 for both methods, as the initial parameters of the models are the same. Then, after 50 epochs, for HYPADE-Fixed, the loss value drops to 45.90, whereas for HYPADE, it decreases to 2.02. The comparatively higher drop in loss is the result of dynamically updating the centroid. After around 500 epochs, for the HYPADE algorithm, the loss value goes below the $loss\_threshold$, and the algorithm terminates. On the contrary, the loss value is 27.13 after 1,000 epochs for HYPADE-Fixed. This demonstrates that using a fixed centroid makes it hard for the model to converge. A similar trend is also evident for the other datasets. For dataset CiteSeer (Figure 2-b), after 1,000 epochs, the loss value decreases to 0.074 for HYPADE-Fixed, whereas for HYPADE, the loss value is 0.008 only. For CoraA, the HYPADE algorithm converges after around 100 epochs, but the loss value for HYPADE-Fixed is 0.113 after 1,000 epochs, which is relatively higher. For dataset Cora (Figure 2-d), HYPADE converges after around 50 epochs, but the loss value is still higher after a thousand epochs.

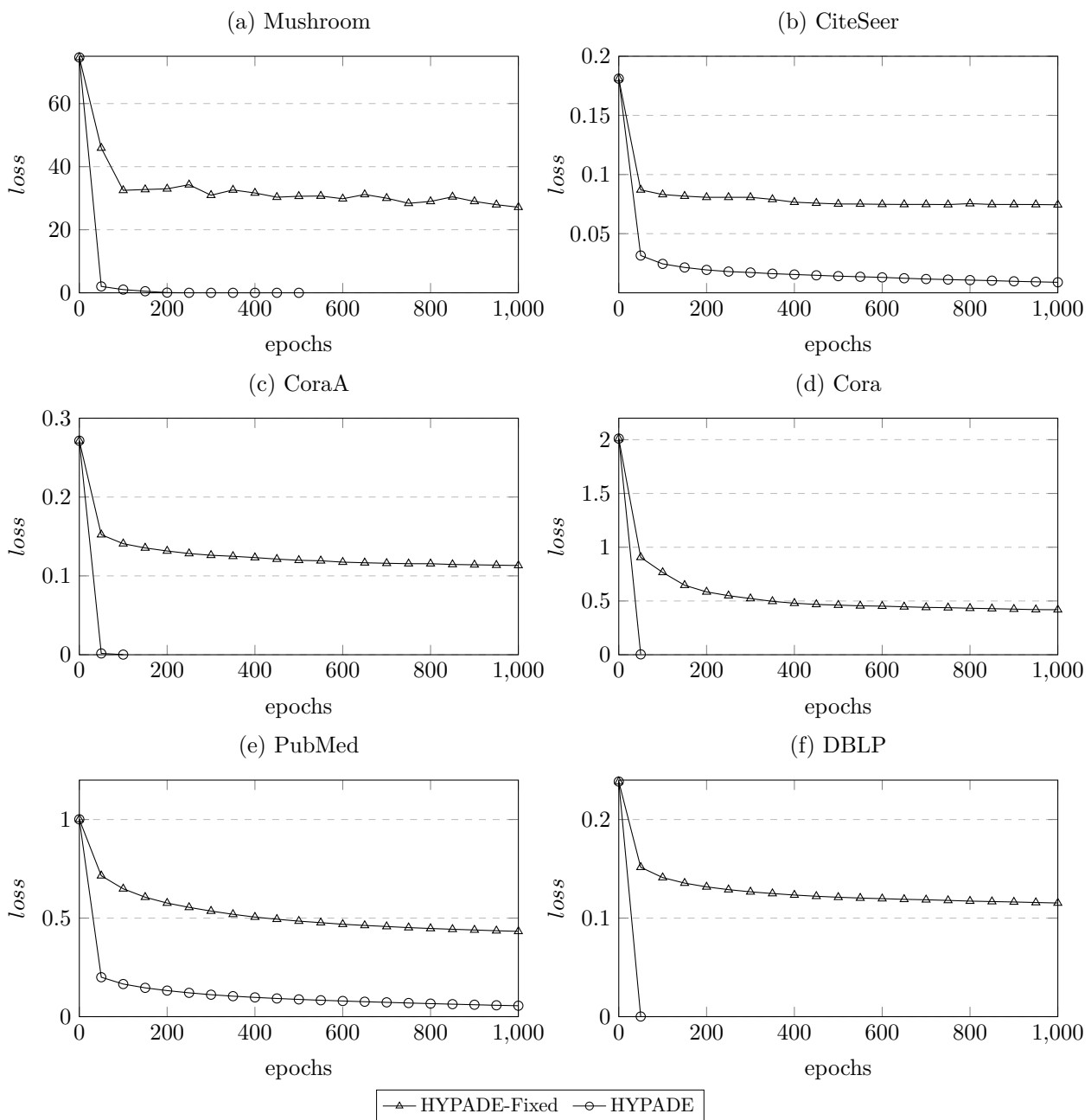

Figure 2: Loss value analysis over epochs for real-life datasets

For dataset PubMed (Figure 2-e), the loss values are 0.432 and 0.055 for HYPADE-Fixed and HYPADE. For dataset DBLP (Figure 2-f), HYPADE converges after around 50 epochs, but the loss value is 0.115 after a thousand epochs.

The loss value analysis over epochs for the synthetic datasets is provided in Appendix A.3.

## 5.5 Visualization

In Figure 3, we visualize the anomaly scores of the hyperedges for the six real-life datasets. In most cases, the anomaly scores for inliers are clustered close to zero, while anomalies tend to have higher scores. In 3-a,

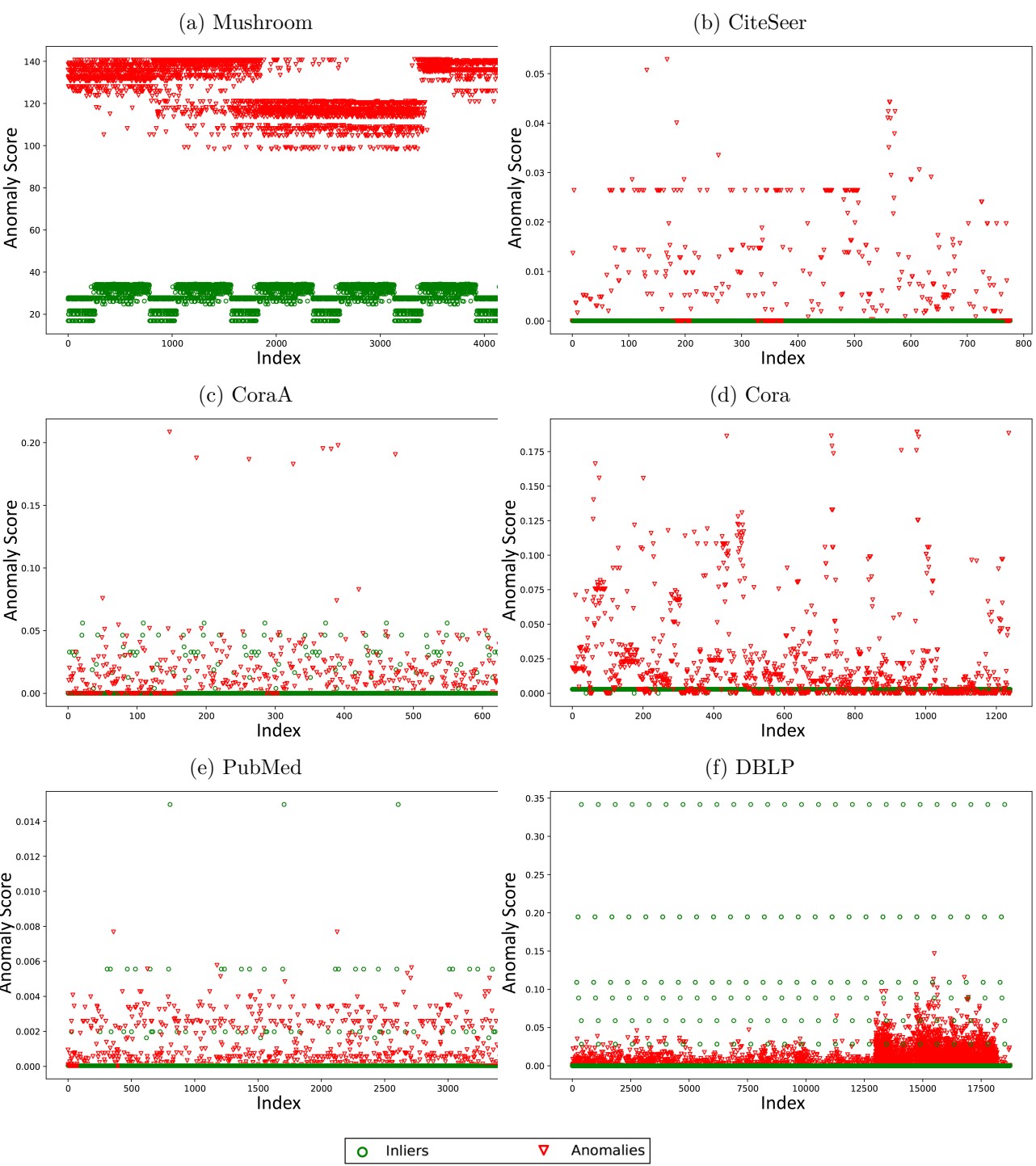

Figure 3: Visualization of anomaly scores for real-life datasets

for the dataset Mushroom, a clear separation is evident between the anomaly scores for the anomalies and inliers. The clear separation of anomaly scores reflects the higher AUROC score of 100% for this dataset. For dataset CiteSeer (Figure 3-b), the anomaly scores are relatively spread out, but anomalies are still distinguishable from inliers. For datasets CoraA and Cora (Figure 3-c and Figure 3-d), inliers are clustered near the bottom, but anomalies are scattered over a wider range of scores. In the case of PubMed (Figure

3-e), anomalies have higher scores, while inliers are more uniformly distributed near zero. In dataset DBLP (Figure 3-f), many anomalous hyperedges have higher scores than inliers. However, there is some overlap near the bottom, and some of the inliers have higher anomaly scores than anomalies. This justifies the relatively lower AUROC score than other datasets.

The visualization of anomaly scores for the synthetic datasets is provided in Appendix A.4.

## 6    Conclusion

In this paper, we have proposed HYPADE, an anomaly detection algorithm for hyperedges in a hypergraph. Our proposed model is unsupervised and needs no labeled data. According to the process, we have devised an end-to-end model that employs a hypergraph neural network to learn hyperedge representations and then predicts the anomaly score with a one-class classifier. To the best of our knowledge, we are the first to propose a deep neural network-based model for anomaly detection on hypergraphs. We have experimented on six real-life hypergraph datasets from different domains and six statistically generated synthetic datasets to evaluate the performance. Finally, we have performed a comparative result analysis of our method against the state-of-the-art research. A significantly higher AUROC score across the datasets demonstrates the effectiveness of our algorithm. Future research directions for this work may include utilizing labeled data, entire anomalous hypergraph detection, etc.

### Acknowledgments

This work is partially supported by (a) University of Dhaka, (b) Natural Sciences and Engineering Research Council of Canada (NSERC) and, (c) University of Manitoba.

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

# A   Synthetic Dataset Experiments

## A.1   Synthetic Datasets

For the synthetic datasets, we have created three hypergraphs with 100 nodes and 3,000 hyperedges. Each node in the hypergraph is initialized with a 128-dimensional feature vector. These feature vectors are randomly generated using values drawn from a uniform distribution over the interval $[0, 1)$. To generate the hyperedges, we followed three different sampling techniques in the three hypergraphs. For one hypergraph, we used uniform sampling on the nodes. Next, we have also applied similarity and dissimilarity-based sampling techniques to generate homophilic and heterophilic associations in the hypergraphs, respectively. The three sampling techniques for hyperedges are presented below in detail.

**Uniform Sampling:** In this approach, each hyperedge is formed by uniformly sampling a random number k of nodes (where $2 \leq k \leq 5$). The selected nodes are grouped to form a hyperedge without any preference or bias. This technique does not require node features and serves as a baseline to simulate random relationships among nodes.

**Similarity-Based Sampling:** This method leverages node features to form semantically coherent hyperedges. For each hyperedge, k candidate nodes are repeatedly sampled, and the average pairwise cosine similarity among their feature vectors is computed. The group with the highest average similarity score is selected to form the hyperedge. This encourages nodes with similar characteristics to be associated together in the hyperedges, modeling homophilic relationships.

**Dissimilarity-Based Sampling:** Contrary to similarity-based sampling, this strategy constructs hyperedges consisting of dissimilar nodes. The one with the lowest average pairwise cosine similarity is chosen among multiple candidate groups. This approach promotes heterophilic connectivity and is useful for modeling diverse or contrasting interactions within the data.

Using the three hypergraph generation techniques, we construct six synthetic datasets for our experiments. In each dataset, one type of hypergraph is treated as the inlier class, while another type serves as the anomaly for measuring the performance. For example, hypergraphs generated using the uniform distribution are considered inliers in one dataset, whereas those generated using similarity-based sampling are treated as anomalies. We present the six synthetic datasets in Table 4.

## A.2   Results on Synthetic Datasets

Table 5 shows the AUROC scores (in percentages) of our model compared to the baseline methods on the synthetic datasets, each simulating different distributions and relationships between inliers and anomalies. While hashing methods such as LSH and HashNWalk show consistently low AUROC scores (mostly below 60%), and VEM performs moderately better, our HYPADE variants clearly outperform them. Among the baseline methods, HGNN consistently achieves stronger performance than the traditional anomaly detection approaches. Among the HYPADE variants, on the dataset Uni-Sim HYPADE-Fixed outperforms HYPADE

Table 4: Synthetic datasets.

| Dataset Name | Hyperedge Sampling | |
| --- | --- | --- |
| | Inlier | Anomaly |
| Uni-Sim | Uniform | Similarity-based |
| Uni-Dis | Uniform | Dissimilarity-based |
| Sim-Uni | Similarity-based | Uniform |
| Sim-Dis | Similarity-based | Dissimilarity-based |
| Dis-Uni | Dissimilarity-based | Uniform |
| Dis-Sim | Dissimilarity-based | Similarity-based |

Table 5: AUROC score (%) of our method and baseline methods on synthetic datasets.

| Dataset | Uni-Sim | Uni-Dis | Sim-Uni | Sim-Dis | Dis-Uni | Dis-Sim |
| --- | --- | --- | --- | --- | --- | --- |
| LSH | 55.37±1.82 | 56.28±1.64 | 47.23±1.91 | 44.12±1.76 | 52.91±1.53 | 52.47±1.68 |
| HashNWalk | 61.32±1.45 | 59.37±1.57 | 54.81±1.83 | 49.26±1.62 | 60.07±1.48 | 59.94±1.71 |
| VEM | 83.43±1.08 | 79.36±1.14 | 59.32±1.66 | 50.11±1.58 | 63.22±1.39 | 59.24±1.51 |
| HGNN | 86.73±0.96 | 84.12±1.05 | 71.84±1.22 | 61.43±1.34 | 80.56±1.08 | 68.21±1.17 |
| HYPADE-Mean | 89.12±0.84 | 52.42±1.73 | 69.31±1.27 | 61.75±1.18 | 36.05±1.81 | 45.26±1.69 |
| HYPADE-Fixed | **90.59**±0.71 | 81.84±0.98 | 67.16±1.41 | 48.55±1.52 | 69.08±1.26 | 53.60±1.44 |
| HYPADE | 89.60±0.76 | **92.86**±0.58 | **78.03**±0.93 | **67.56**±1.01 | **92.03**±0.62 | **73.56**±0.88 |

by a slight margin. On all five other datasets, HYPADE performs the best, which again proves the effectiveness of utilizing *maxmin* pooling and dynamic centroid updating in our proposed algorithm. On dataset Uni-Dis, HYPADE achieves a high AUROC score of 92.86%. A similar high AUROC score of 92.03% is obtained for Dis-Uni also. For datasets Sim-Uni, Sim-Dis, and Dis-Sim, the AUROC scores are moderately high.

## A.3 Loss Analysis on Synthetic Datasets

In Figure 4, we illustrate the loss progression over the first 100 epochs for the synthetic datasets. Similar to real-life datasets, we compare HYPADE-Fixed with our proposed HYPADE algorithm to highlight the impact of dynamically updating the centroid during training. A similar trend to real-life datasets is also evident in the loss values over epochs for synthetic datasets. Across all datasets, both models exhibit a sharp reduction in loss during the early epochs, followed by a gradual decrease in loss minimization rate. However, the initial steeper decline in loss facilitates the proposed HYPADE model to converge faster, contributing to its superior performance. Our model aims to map the hyperedge embeddings closely, where the hyperedge embeddings are determined based on the similarity of the constituent nodes. Therefore, it becomes easier for the model to learn the mapping when the degree of similarity among the nodes in a hyperedge is similar, whether it be homophilic or heterophilic. For datasets Sim-Uni, Sim-Dis, Dis-Uni, and Dis-Sim, inliers are sampled based on node similarity. As a result, compared to the other two datasets, Uni-Sim and Uni-Dis, the model converged in a relatively lower number of epochs. In contrast, for the Uni-Sim and Uni-Dis datasets, inliers are sampled from a uniform distribution instead of being grouped based on similarity. In consequence, the model requires a relatively higher number of epochs to converge.

## A.4 Visualization of Anomaly Scores for Synthetic Datasets

We illustrate the distribution of anomaly scores for hyperedges in the six synthetic datasets in Figure 5. Similar to the real-life datasets, the tendency for the anomaly scores for inliers to be close to zero is evident in the synthetic datasets. In contrast, the anomalous hyperedges are assigned higher scores. In 5-a, for the dataset Uni-Sim, it is evident that the inliers have anomaly scores close to zero. In contrast, for the

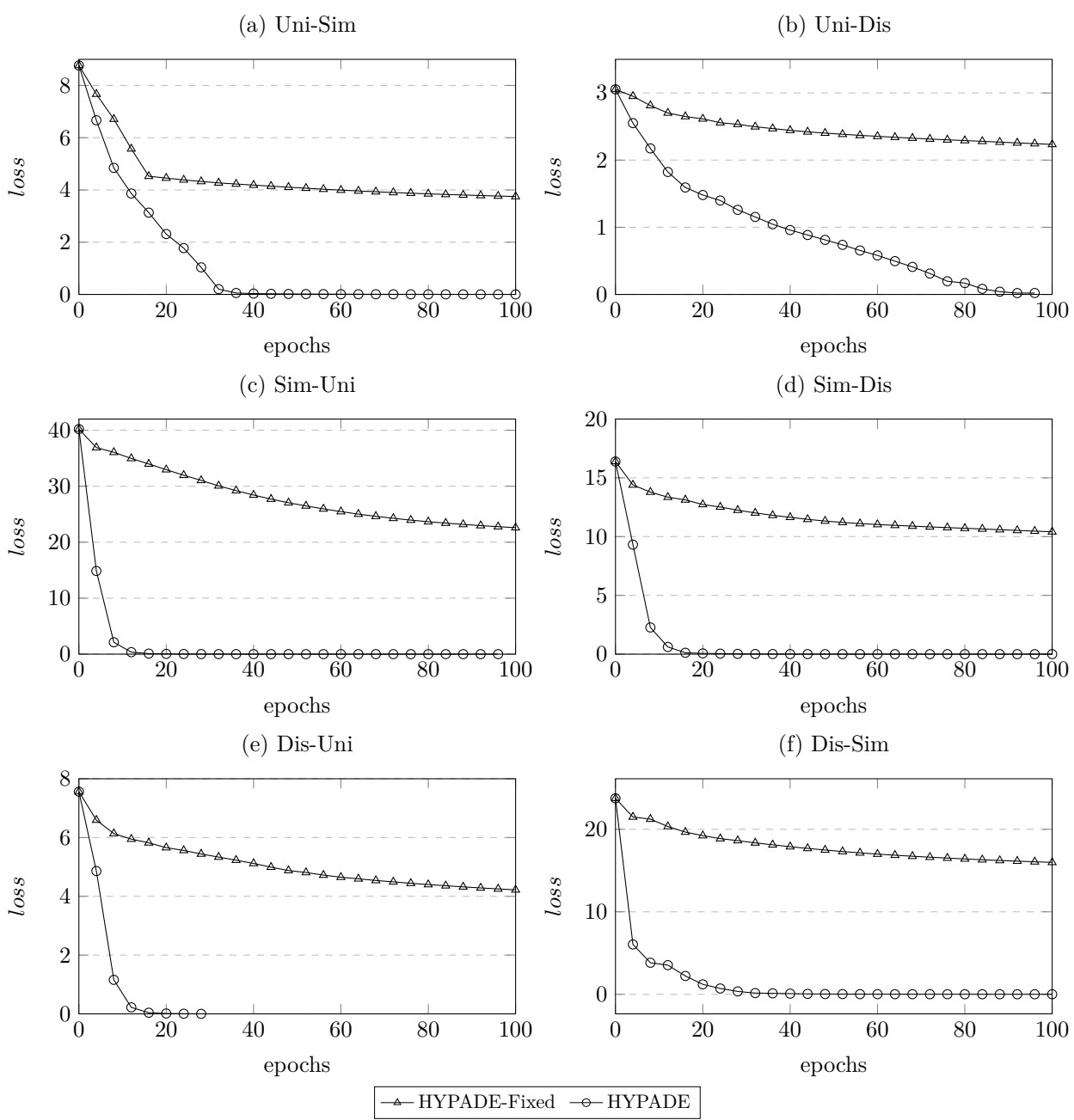

Figure 4: Loss value analysis over epochs on synthetic datasets

anomalies, anomaly scores are distributed with some overlaps but tend to have higher anomaly scores, forming a distinct upper band. For dataset Uni-Dis (Figure 5-b), a similar pattern is visible. The anomaly scores for anomalous hyperedges are spread out, whereas for inliers, the scores are clustered towards zero. The clear distinguishability in the patterns of anomaly scores between inliers and anomalies reflects the relatively higher AUROC score for these two datasets. For datasets Sim-Uni (Figure 5-c), between anomaly scores 0 and 20, the inliers and anomalies can be easily distinguishable. However, some inliers are seen to be assigned anomaly scores much higher than those of anomalies contributing to the lower AUROC score. For the Sim-Dis dataset (Figure 5-d), the anomaly scores of inliers and outliers show significant overlap, making it difficult to distinguish between the two groups. This lack of separation corresponds to the relatively low AUROC score of just 67.56%. In the case of Dis-Uni (Figure 5-e), the inliers and anomalies are easily

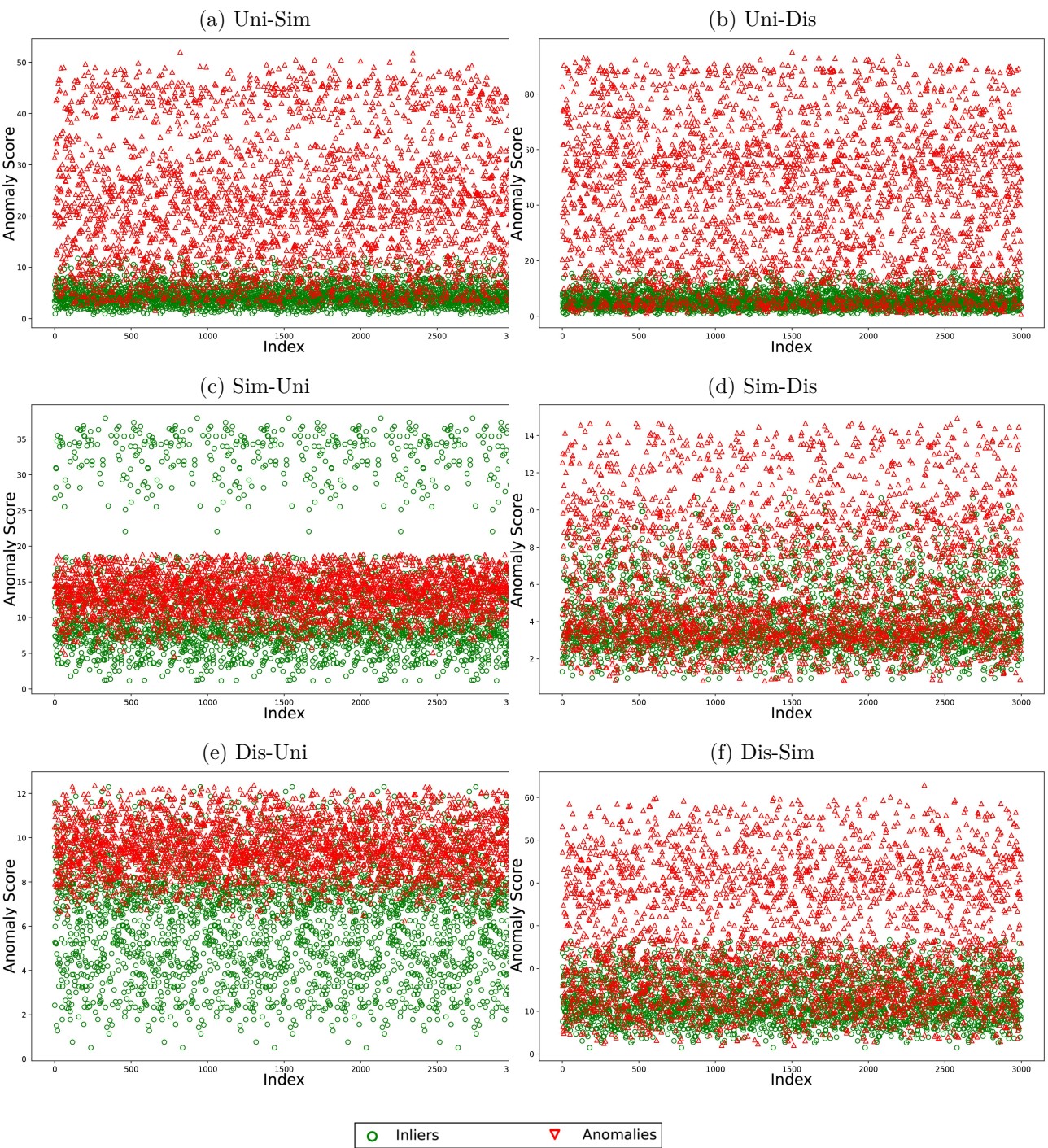

Figure 5: Visualization of anomaly scores for synthetic datasets

separable with a few overlaps leading to a higher AUROC score of 92.03%. In dataset Dis-Sim (Figure 5-f), there is visible score stratification, with anomalies above inliers, but some outliers are mapped to lower scores like inliers.

