# OpenReview forum: "Hyperedge Anomaly Detection with Hypergraph Neural Network"
_TMLR — Decision pending for TMLR_

### Review · Reviewer_Ek1a · 2026-04-07

**Summary Of Contributions:**

The paper proposes HYPADE, an unsupervised model for detecting anomalous hyperedges in attributed hypergraphs. It uses a two-stage AllSet-style message passing, then produces final hyperedge embeddings via a maxmin pooling intended to capture intra-hyperedge diversity.

**Strengths**

1. Hyperedge-level anomaly detection with deep models is under-explored, this paper fills this gap with HYPADE

2. Method is simple and easy to implement.

3. The evaluation includes ablations on pooling and centroid, and controlled synthetic data.

**Weaknesses**

1. The supervised variant seems never appeared in the paper.

2. No deep baselines such as HGNN, HyperGCN,  it is hard to attribute gains to the proposed components.

3. The hypersphere collapse mitigation is a heuristic with one global threshold across all datasets.

**Audience:**

Yes

**Audience Explanation:**

This topic of Anomaly Detection with hypergraph will arouse interest of some TMLR audience.

**Broader Impact Concerns:**

No Broader Impact Statement is included in this paper.

**Claims And Evidence:**

Yes

**Claims Explanation:**

Yes, experiments are provided to support the claim.

**Requested Changes:**

1. Please explain the supervised variant mentioned in the abstract.

2. Please add a deep hypergraph baseline: HGNN or AllSet as backbone with the same one-class loss.

3. Please state explicitly what is borrowed from AllSet vs new,  the two-stage update looks quite similar.

---

> ### Author Response · Authors · 2026-06-13
> **Response to requested change 1:**
>
> We thank the reviewer for pointing out this ambiguity. The current manuscript only presents and evaluates the unsupervised formulation of HYPADE. Although the framework could potentially be extended to a supervised setting by incorporating labeled anomalies into the objective function, such a formulation was not developed or experimentally evaluated in the submitted version. To avoid confusion, we have removed the statement claiming that the model operates in both supervised and unsupervised settings and revised the manuscript to consistently describe HYPADE as an unsupervised hyperedge anomaly detection framework.

---

> ### Author Response · Authors · 2026-06-13
> **Response to requested change 2:**
>
> We appreciate this suggestion. Following the reviewer's recommendation, we have added HGNN as an additional deep hypergraph baseline. We also note that HYPADE-Mean is equivalent to an AllSet-style mean aggregation encoder combined with our one-class anomaly detection objective. The new experimental results and corresponding discussion have been included in the revised manuscript.

---

> ### Author Response · Authors · 2026-06-13
> **Response to requested change 3:**
>
> We thank the reviewer for this observation and have clarified the relationship between our method and AllSet in the revised manuscript. Like AllSet, our model employs a two-stage message-passing mechanism in which information is first aggregated from nodes to hyperedges and then from hyperedges back to nodes. However, the primary contribution of HYPADE does not lie in proposing a new message-passing architecture. Instead, our novelty arises from adapting this framework to the previously unexplored problem of hyperedge anomaly detection and introducing a dedicated anomaly detection objective based on hyperedge representations.
> More specifically, HYPADE differs from AllSet in three important aspects: (1) it learns hyperedge embeddings specifically for anomaly detection rather than prediction or classification tasks; (2) it introduces a max–min pooling mechanism to capture diversity among nodes within a hyperedge when constructing hyperedge representations; and (3) it employs a one-class optimization objective with a dynamically updated centroid to assign anomaly scores to hyperedges.

---

### Review · Reviewer_XYvK · 2026-04-10

**Summary Of Contributions:**

The paper proposes HYPADE, an end-to-end hypergraph neural network for hyperedge anomaly detection. The method learns node embeddings through alternating node and hyperedge updates, builds final hyperedge embeddings with max-min pooling, and scores anomalies by distance to a learned hypergraph centroid in a one-class objective. The empirical study includes six real-world datasets and six synthetic datasets, with comparisons to three prior hyperedge-anomaly baselines plus two internal ablations.

Strengths and weaknesses:
- Strength: The paper tackles a relatively underexplored problem, hyperedge anomaly detection, and presents a simple, end-to-end formulation that is easy to follow.
- Strength: The experimental section includes both real and synthetic datasets, and the ablations on mean pooling versus max-min pooling and fixed versus dynamic centroid are useful.
- Weakness: The evaluation setup raises concerns about realism and fairness. Several anomaly labels are constructed heuristically from class-label frequency in citation and co-authorship datasets, which is only a proxy for anomaly detection, and the paper does not clearly justify this choice.
- Weakness: The strongest claims are not fully supported, because comparisons are limited to older baselines, variance or significance is not reported in the main tables, and one stated contribution, operation in both unsupervised and supervised settings, is not actually developed or evaluated in the paper.

**Audience:**

Yes

**Audience Explanation:**

Hypergraph anomaly detection is an interesting and comparatively less explored problem, and a neural approach for scoring anomalous hyperedges is relevant to readers interested in graph learning, anomaly detection, and higher-order relational data. Even if the current evaluation is not yet fully convincing, the problem formulation and the proposed direction are likely to interest part of the TMLR audience.

**Broader Impact Concerns:**

None.

**Claims And Evidence:**

No

**Claims Explanation:**

The paper provides encouraging evidence, especially the ablations and the results in Table 3, but the support is not fully convincing for the broader claims. The benchmark construction is partly artificial, the baseline set is narrow, improvements are sometimes marginal rather than substantial, the main results do not report uncertainty despite five runs being mentioned, and the paper claims supervised capability without presenting a corresponding method or experiments.

**Requested Changes:**

- Critical: Strengthen the evaluation protocol. Clarify exactly how anomalies are defined in each dataset, justify why these proxy labels correspond to anomaly detection, and discuss limitations of this setup.
- Critical: Add stronger baselines, including simple feature-based or non-neural one-class methods and, if possible, more competitive hypergraph representation baselines that use node features.
- Critical: Report mean with standard deviation or confidence intervals, and include significance testing where appropriate. The paper states five runs were used, but the tables only show point estimates.
- Critical: Resolve the inconsistency around the claim that the model operates in both unsupervised and supervised settings, either by adding the supervised formulation and experiments or by removing that claim.
- Strengthening: Add sensitivity analyses for the loss threshold, centroid update choice, and architectural depth, since the collapse-avoidance mechanism currently appears heuristic.

---

> ### Author Response · Authors · 2026-06-13
> **Response to requested change 1:**
>
> We thank the reviewer for this valuable suggestion. We have revised Section 5.2.1 to provide a clearer description of how anomaly labels are defined for each dataset. For the Mushroom dataset, hyperedges corresponding to poisonous mushroom species are treated as anomalies, while edible species are treated as normal instances. For the citation and co-authorship datasets (CiteSeer, Cora, CoraA, PubMed, and DBLP), we follow the commonly adopted hyperedge labeling strategy in prior hypergraph learning studies, where hyperedge labels are derived from the labels of their constituent nodes. Specifically, we identify the most frequent class in each dataset and treat hyperedges associated with that class as inliers, while hyperedges associated with all other classes are treated as anomalies.
> To address the reviewer's concern regarding the validity of these labels, we have added a discussion in Section 5.2.1 clarifying that the anomaly labels are proxy labels rather than naturally occurring anomaly annotations. We explain that publicly available benchmark datasets with ground-truth hyperedge anomaly labels are currently scarce, necessitating the use of a proxy-labeling strategy to enable quantitative evaluation. We further acknowledge the limitations of this setup by noting that class membership does not necessarily correspond to real-world anomalous behavior. This discussion has been added explicitly to improve the transparency of the evaluation protocol and to clarify the scope and limitations of our experimental findings.

---

> ### Author Response · Authors · 2026-06-13
> **Response to requested change 2:**
>
> We appreciate this suggestion and agree that additional baselines strengthen the empirical evaluation. In the revised manuscript, we have included additional baselines in Tables 3 and 5.

---

> ### Author Response · Authors · 2026-06-13
> **Response to requested change 3:**
>
> We thank the reviewer for identifying this omission. The original experiments were repeated across five runs; however, only the mean AUROC values were reported. In the revised manuscript, all performance tables have been updated to report mean ± standard deviation over the five runs.

---

> ### Author Response · Authors · 2026-06-13
> **Response to requested change 4:**
>
> We agree with the reviewer. The current manuscript only presents and evaluates the unsupervised formulation of HYPADE. Although the framework could potentially be extended to a supervised setting by incorporating labeled anomalies into the objective function, such a formulation was not developed or experimentally evaluated in the submitted version.
> To avoid confusion, we have removed the statement claiming that the model operates in both supervised and unsupervised settings and revised the manuscript to consistently describe HYPADE as an unsupervised hyperedge anomaly detection framework.

---

> ### Author Response · Authors · 2026-06-13
> **Response to requested change 5:**
>
> We appreciate this important suggestion. In the current manuscript, the evolution of the loss value over training epochs is presented, illustrating the stability and convergence behavior of the proposed optimization process. In addition, the centroid update strategy is explicitly described and evaluated through a comparison between fixed-centroid and dynamic-centroid optimization mechanisms, with anomaly detection accuracy reported for both settings. This comparison demonstrates the effectiveness of the proposed dynamic centroid optimization mechanism in preventing representation collapse and improving anomaly discrimination.

---

### Review · Reviewer_ifpx · 2026-05-17

**Summary Of Contributions:**

This paper investigates the topic of anomaly detection in hypergraphs. To this end, the authors present HYPADE, an end-to-end hypergraph neural network framework that learns node and hyperedge representations through two-stage node–hyperedge message passing. HYPADE constructs final hyperedge embeddings using a max–min pooling operator, and then assigns anomaly scores based on the Euclidean distance between each hyperedge embedding and a dynamically updated hypergraph centroid. Experimental results may demonstrate the effectiveness of the proposed HYPADE.

**Additional Comments:**

NIL.

**Audience:**

Yes

**Audience Explanation:**

Hypergraph anomaly detection investigated in this paper is relevant to TMLR because it is an underexplored problem. And using hypergraph neural networks for this task is potentially valuable.

**Broader Impact Concerns:**

Not applicable for this paper.

**Claims And Evidence:**

Yes

**Claims Explanation:**

The claims made in the submission are partially supported by the results presented in the paper. The presented results indicate that HYPADE can outperform existing hyperedge anomaly detection baselines in several cases, especially on Mushroom, CiteSeer, CoraA, Cora, and most synthetic settings. However, there are some issues that require further justification.
1. The term anomalous hyperedges needs further justification. For example, in citation and co-author datasets, hyperedges are labeled as inliers or anomalies based on whether they contain nodes with the most frequent node label. Such a labeling method may result in class imbalance or topic/category differences rather than genuine hyperedge anomalies. The subsequent evaluation may not fully validate the claimed ability to detect abnormal higher-order associations.
2. To make the comparisons fair, the authors are suggested to include more hypergraph neural encoders integrated with the anomaly scoring objective, similar to the proposed HYPADE, in the experiments.
3. A more sufficient sensitivity analysis, and embedding variance analysis should be conducted to discuss how the issue of representation collapse is addressed by the proposed method. HYPADE uses a predefined loss threshold to solve the issue. It looks like a heuristic approach and requires more empirical results to show that the learned representations under this “loss threshold” are meaningful.

**Requested Changes:**

1. The anomaly construction protocol for real-world datasets is not sufficiently convincing. For example, in citation and co-author datasets, hyperedges are labeled as inliers or anomalies based on whether they contain nodes with the most frequent node label. Such a labeling method may result in class imbalance or topic/category differences rather than genuine hyperedge anomalies. The subsequent evaluation may not fully validate the claimed ability to detect abnormal higher-order associations.
2. The training objective looks like a heuristic one. HYPADE uses a predefined loss threshold to solve the issue. It looks like a heuristic approach and requires more empirical results to show that the learned representations under this “loss threshold” are meaningful.
3. The novelty and contributions compared with existing hypergraph neural networks and one-class objectives need further clarification. The two-stage message passing strategy adopted by the proposed HYPADE is also (partially) considered by previous approaches such as AllSet and HCoN. And the proposed anomaly scoring module is also similar to one-class classification or other distance-to-center objectives. Thus, the novelty of this paper is more like an engineering combination, which is acceptable. However, the authors may overstate the novelty in the manuscript. The authors are suggested to further clarify the novelty and contributions, or tone down their statement.
4. The experiments can be redesigned to make the comparisons fair. The authors are suggested to include more hypergraph neural encoders integrated with the anomaly scoring objective, similar to the proposed HYPADE, in the experiments.
5. All configurations and settings of all approaches and test datasets should be introduced in the paper. The authors are suggested to prepare a subsection on reproducibility in the manuscript to present all the mentioned details.

---

> ### Author Response · Authors · 2026-06-13
> **Response to requested change 1:**
>
> We appreciate the reviewer's concern regarding the anomaly construction strategy. Since publicly available benchmark datasets with ground-truth hyperedge anomaly labels are currently scarce, we followed the hyperedge labeling strategy adopted in prior hypergraph classification studies [Wu et al. (2023)], where hyperedge labels are derived from node labels when explicit hyperedge labels are unavailable. Specifically, for the citation and co-authorship datasets, we considered hyperedges containing nodes from the majority class as inliers and the remaining hyperedges as anomalies.
> We agree that this protocol does not perfectly capture all forms of anomalous higher-order associations and may partially reflect topic heterogeneity. To mitigate this limitation, we additionally evaluated HYPADE on six synthetic datasets specifically designed to model different structural and feature-based distributions between normal and anomalous hyperedges (Section A2). Furthermore, we included the Mushroom dataset, which provides original class labels and therefore does not require any artificial anomaly construction procedure. The consistent performance improvements observed across both real-world and synthetic datasets suggest that HYPADE can capture meaningful higher-order patterns beyond simple label-frequency effects.

---

> ### Author Response · Authors · 2026-06-13
> **Response to requested change 2:**
>
> We thank the reviewer for highlighting this point. The loss threshold was introduced to address the hypersphere collapse issue that arises when the centroid is updated dynamically during optimization. Similar collapse phenomena have been discussed in prior one-class learning literature. In our framework, the threshold acts as a stopping criterion that prevents the trivial solution where all embeddings collapse to a single point. Because of dynamically updating the centroid during optimization, as shown in Figure 2, the loss value exhibits a characteristic evolution when the centroid is updated iteratively: after an initial decrease, the loss can continue shrinking toward a trivial solution as the centroid adapts to the embeddings. Figure 4 further analyzes this behavior and illustrates how dynamic centroid updates influence the optimization trajectory. These results highlight the collapse tendency induced by centroid adaptation and motivate the need for a mechanism to prevent the optimization process from converging to degenerate representations.
> Regarding the meaningfulness of the learned representations, Figures 3 and 5 provide embedding visualizations showing that normal and anomalous hyperedges remain well separated in the learned latent space. These visualizations support our claim that the representations learned before reaching the threshold are discriminative and capture meaningful structural patterns. We will clarify these points in our response and emphasize that the threshold is used solely to prevent hypersphere collapse during training and is not involved in the anomaly scoring function itself.

---

> ### Author Response · Authors · 2026-06-13
> **Response to requested change 3:**
>
> We appreciate this observation and agree that the original manuscript may have overstated certain novelty claims.
> Our intention is not to claim novelty for the individual components in isolation. We acknowledge that the node-hyperedge message passing mechanism is related to prior hypergraph neural network architectures such as AllSet and HCoN, while the distance-to-center objective is inspired by established one-class learning methods.
> The primary contribution of HYPADE lies in the integration of these components into a unified end-to-end framework specifically designed for hyperedge anomaly detection, a task that has received very limited attention in the literature. In particular, our framework introduces:
> 1. A hyperedge-focused representation learning mechanism that explicitly constructs hyperedge embeddings for anomaly detection rather than classification or prediction tasks.
> 2. A max-min pooling strategy that captures diversity among constituent nodes within a hyperedge.
> 3. A dynamically updated centroid-based one-class objective tailored to hyperedge anomaly detection.

---

> ### Author Response · Authors · 2026-06-13
> **Response to requested change 4:**
>
> We thank the reviewer for this valuable suggestion. We agree that comparing against additional hypergraph neural network encoders combined with a common anomaly scoring objective would provide a stronger evaluation of the representation learning component.
> In the revised version, we will extend the experimental study by incorporating additional baselines (Tables 3 and 5). We believe these additional comparisons will provide a more comprehensive and fair assessment of the proposed framework.

---

> ### Author Response · Authors · 2026-06-13
> **Response to requested change 5:**
>
> We agree with the reviewer and have extended the experimental setup section with the implementation details necessary to reproduce the experiments. Hence, we add Section 5.1 on Experimental Setup.

---

### Decision · Action_Editor_MH71 · 2026-06-24

**Recommendation:** Accept as is

**Audience:**

Yes

**Audience Explanation:**

Hypergraph anomaly detection is an underexplored problem that should be of interest to many members of the TMLR audience.

**Claims And Evidence:**

Yes

**Claims Explanation:**

The paper studies the problem of anomaly detection in hypergraphs and proposes an unsupervised approach for the task, called HYPADE, based on hypergraph neural networks. HYPADE learns node and hyperedge representations and assigns anomaly scores based on the Euclidean distance between hyperedge embeddings and a hypergraph centroid. Results are presented on several real-world and synthetic datasets, with comparisons to baselines and ablation studies. The empirical results support the main claims in the paper about the proposed algorithm.

Initially, reviewers raised several issues, such as unsupported mentions of a supervised version of the algorithm, concerns about the way anomalies are defined in real-world datasets and the absence of certain baselines for the experimental results. The authors addressed all the concerns raised by the reviewers, strengthening the paper, and adjusting the claims in the paper accordingly.